# Numerical Modeling and Simulation of the Effectiveness of Groundwater Source Protection Management Plans: Riverbank Filtration Case Study in Serbia

**Dušan Polomčić [1], Dragoljub Bajić [1,*](ID), Bojan Hajdin [1] and Dragan Pamučar [2](ID)**

[1] Faculty of Mining and Geology, University of Belgrade, 11000 Belgrade, Serbia; dusan.polomcic@rgf.bg.ac.rs (D.P.); bojan.hajdin@rgf.bg.ac.rs (B.H.)

[2] Department of Logistic, University of Defense in Belgrade, 11000 Belgrade, Serbia; dragan.pamucar@va.mod.gov.rs

[*] Correspondence: dragoljub.bajic@rgf.bg.ac.rs

**Abstract:** The paper describes the establishment and testing of an algorithm for developing sustainable management plans associated with groundwater source protection against potential pollutants and discusses the effectiveness of individual systems. The applied methodology pertains to groundwater resource management, particularly those cases that involve riverbank filtration. Namely, groundwater (numerical) modeling is employed to examine the groundwater regime and balance, as well as to create protection systems and illustrate their effectiveness. Particle tracking analysis is used to study pollutants' travel and residence time. On the other hand, PEST with regularization is employed to estimate the numerical model parameters. The proposed method is used in a real case study, which examines the application of the developed algorithm to the protection of a drinking water supply source from an industrial zone, which is a potential source of pollution. The research presented in the paper opens new avenues for future studies involving mathematical multicriteria optimization and decision making about optimal groundwater source protection management plans.

**Keywords:** groundwater resource management; groundwater modeling; PEST with regularization; particle tracking analysis; pollutant





## 1. Introduction

Groundwater quality is a key factor when groundwater is used for various purposes, especially as a source of drinking water. In developing countries, this resource is increasingly threatened by a growing number of pollution sources and pollutants. In general, the origin of such pollutants is either natural (geogenetic), where rocks dissolve in contact with groundwater, or due to intruding saltwater or other water of poor quality, or, more often, anthropogenic [1]. The US EPA [2] classifies groundwater pollution sources under six categories, depending on how the pollutants are released. [3] also proposes six categories of pollution sources, based on the origin of the pollutants. The pollutants can roughly be grouped into chemical, biological and radioactive [4]. Nitrogen compounds (nitrates, nitrites, ammonia) are among the most frequent inorganic pollutants found in groundwater. They are generally indicative of anthropogenic sources, because such compounds are often used in agriculture, but they are also found in household wastewater [5–7].

Heavy metals can be a major groundwater quality issue where groundwater is used for the drinking water supply. If found in concentrations higher than those permitted by regulations, they can be harmful to human health, and, at times, even carcinogenic. For example, this is the case with arsenic, chromium, mercury and cadmium [8,9]. Elevated concentrations of arsenic have been recorded in Vietnam, in an anoxic aquifer rich in iron [10]. Cases of groundwater pollution with cadmium have been reported in India [11,12].

Industrial progress has led to an increasing demand for organic compounds. Even those that are not of natural origin, such as pesticides, pharmaceuticals, hormones, petroleum products, and other similar substances, can be detected in groundwater [1,13,14]. Because of their widespread use, petroleum products are among the most common pollutants found in groundwater [15]. When the petroleum hydrocarbons reach the environment, their occurrence and fate vary, depending on the composition and physicochemical properties. If their molecular mass is high, they are generally very toxic. Because of this toxic nature, as well as the mutagenic and carcinogenic properties, various procedures are used to remove petroleum hydrocarbons from the environment [15]. Groundwater pollution with petroleum hydrocarbons is usually accidental, as was the case in Nigeria [16], where oil spilled from a damaged pipeline and polluted the environment (including the groundwater). Elevated hydrocarbon concentrations have also been detected in Serbia, on the site of a district heating plant in New Belgrade [17]. In this case, a suspected environmental incident led to testing of the groundwater quality. Samples were collected from ten observation wells. Surface water samples from the Sava River were also analyzed. The results showed the presence of diesel and heavy heating oil, so the site had to be bioremediated to effectively reduce the total hydrocarbon concentrations to acceptable levels. An instance of karst aquifer pollution with petroleum products was reported in China, where the results of a study indicated that the hydrogeologic conditions and biodegradation were the main causes of this phenomenon [18]. The Quaternary sediments were thin and the limestones exposed, contributing to hydrocarbon intrusion into the water-bearing horizon. The karst conduits and caverns governed the migration through the aquifer (convection), and igneous rocks constituted a barrier to the north. However, biodegradation reduces petroleum hydrocarbon concentrations. Giadom and Tse [19] describe the existing and potential adverse effects of the hydrocarbon spills in Nigeria, where the depth to groundwater can be less than 1 m during the heaviest rainfalls. Remediation has lowered the concentrations, but they still exceed the legally permissible levels. In general, different approaches are followed worldwide to deal with problems of this type [20–23].

In view of the large number of potential pollutants, it is extremely important to prevent groundwater pollution or at least provide adequate protection. Numerical modeling is a useful tool which, in synergy with geological, hydrological and hydrogeological methods, can help to predict and prevent the spread of groundwater pollution, especially in the case of groundwater sources used for a public water supply. MODFLOW [24] is a computer program commonly used to model the groundwater regime and/or the processes that take place in the porous environment. There are several graphic interfaces for hydrodynamic modeling, among which Groundwater Vistas [25] is used most often. MODFLOW is based on solving the equations that describe groundwater flow, such that the model it produces is the basis for simulating processes associated with groundwater flow (e.g., contamination). Another notable code is MODPATH [26], which is used for particle tracking analyses. As such, it is useful in hydrogeology for the protection zoning of groundwater sources. One of the most interesting cases in which these two models were applied is a comparative analysis of the methods used for the protection zoning of an unconfined aquifer in Israel [27]. A study was conducted in Morocco on the groundwater source protection zoning, based on the MODFLOW and MODPATH models [28]. MODFLOW was also used for the protection zoning of a groundwater source in Požaranje, Kosovo [29]. For the Njafabad aquifer in Iran, a MODFLOW and WhAEM model were constructed to delineate protection zones for 2.5 and 10 years [30]. In addition, the effect of variable well-pumping rates on the delineated zones was analyzed. MODPATH was used to define the protection zones of four selected wells. The results showed that the zones identified by the WhAEM model were smaller than those indicated by MODFLOW. This might pose a certain risk, especially upstream of the well site. A study conducted in Greece [31] considered the effects of potential point sources of pollution on the groundwater quality in Nea Moundania. The pollution sources were identified first, then the groundwater flow was simulated with a three-dimensional MODFLOW model of finite differences, and finally the protection zones

were determined by particle tracking using MODPATH. A similar approach was followed in Iran for the Iranshahr aquifer, where a model was constructed in MODFLOW, and then the particle travel time to the wells was monitored. The longest travel time was 509 days and the shortest was 144 days [32].

In a nutshell, various methodologies have been used to address the issue of groundwater contamination and groundwater source pollution. The initial assumptions governing the present research stemmed from the facts associated with the effectiveness of numerical modeling when designing groundwater source-protection management plans. A series of potential or suitable plans are provided and the doors opened for tradeoffs between wishes (criteria) and possibilities (constraints), followed by the selection of the optimal plan.

This paper presents a methodology for investigations associated with establishing a groundwater source for public water supply based on riverbank filtration, including the quantification of the environmental impact of pumping, assessment of the influence of an industrial zone, and a hydrodynamic analysis of the effectiveness of three alternative ways of protecting the groundwater source against pollution from the industrial zone. The numerous calculations and analyses involved iterative solving of partial differential equations that described the groundwater flow and simulating the migration path and velocity of a conservative particle along the streamline.

## 2. Study Area

The groundwater source of Jagodica encompasses a portion of the alluvial plain of the Danube, where the ground elevations are generally 70–72 m (Figure 1). The alluvial plain is composed of sands, sandy gravels and gravels, and the strata that overlie the aquifer are comprised of silty sediments and clays. The spread is continuous in the extended zone of Jagodica. The sandy part of the aquifer is in direct hydraulic contact with the Danube River and one of its arms—the Dunavac. The sandy layer overlies gravels (Figure 2). The plans call for groundwater extraction from the aquifer formed in these gravels and sands. The thickness of the alluvial sediments is 17 to 23 m. Pliocene marly clays are the alluvial bedrock.

The water table of the alluvial aquifer is affected by the varying stages of the precipitation and pumping station, Kolište II (Figure 1). The pumping station is located on a downstream dam on the regulated reach of the Dunavac, from where water is pumped immediately upstream of the dam, into the part of the Dunavac which is in contact with the Danube River (Figure 1). The pumping station controls the water levels of the Dunavac and the drainage canals DK-1 and DK-2.2, which, along with 35 drainage wells, protect the area from high groundwater levels. The discharging and charging of the Iron Gates (Đerdap) reservoir, some 200 km downstream from the study area, have the most pronounced effect on groundwater levels.

The plans call for the Jagodica source to be located along the Danube, in the northeastern part of the study area (Figure 1), and to provide a water supply to the city of Požarevac and nearby villages. The source will rely on riverbank filtration and will be comprised of 20 pumping wells, individual capacity 20 L/s, which will extract groundwater from the sandy and gravelly layers of the alluvial aquifer. There is an industrial zone some 800 m west of the groundwater source site, where there are five facilities whose operation can contaminate the source (W1 to W5).

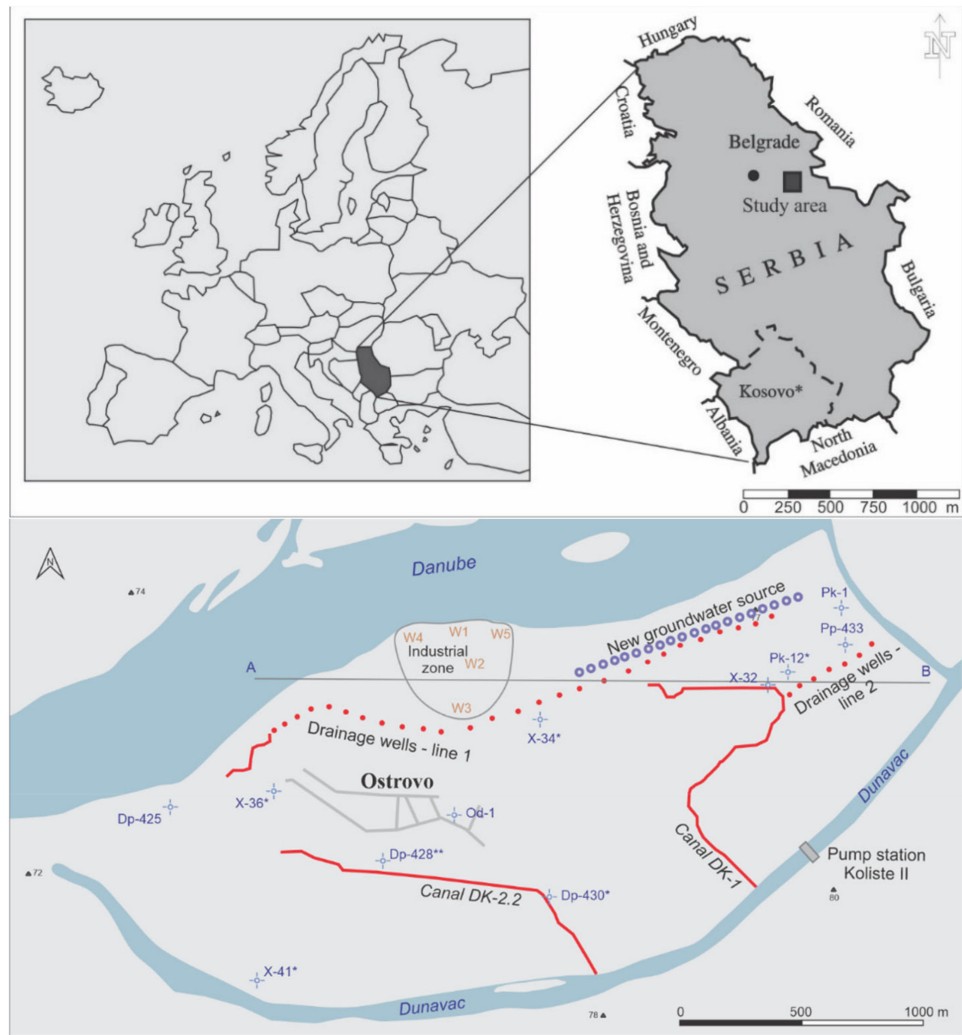

**Figure 1.** Location map of the study area.

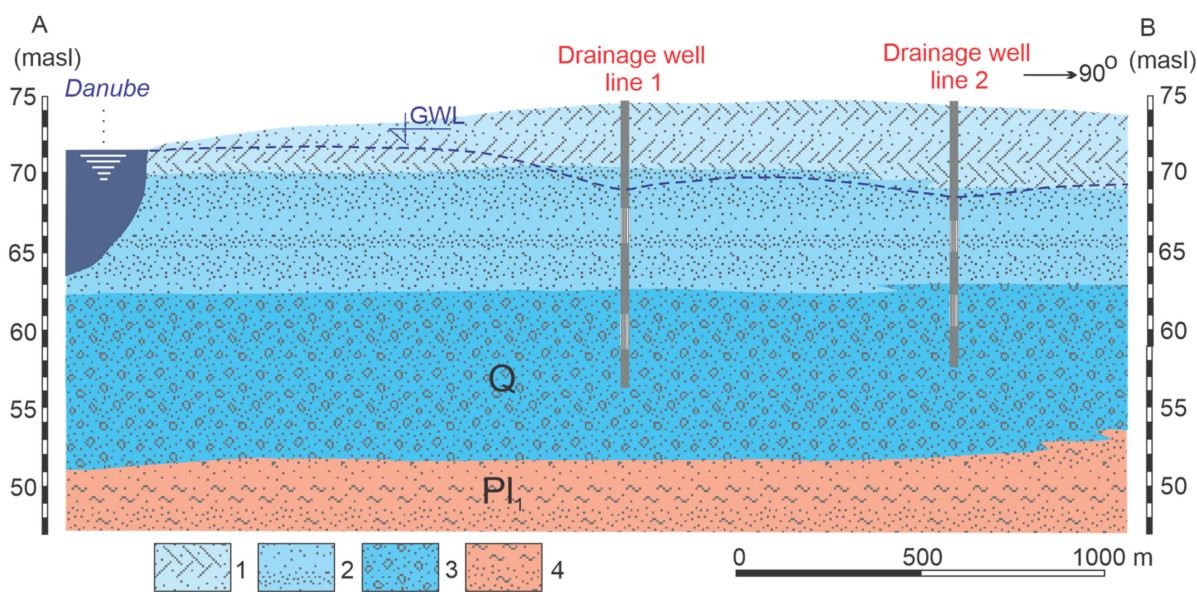

**Figure 2.** Hydrogeological cross-section A–B (Figure 1). Legend: 1. Sandy clay; 2. Sand; 3. Gravel; 4. Sandy clay, clay.

## 3. Methodology

Figure 3 shows an algorithm developed in this study, which generally comprises seven stages.

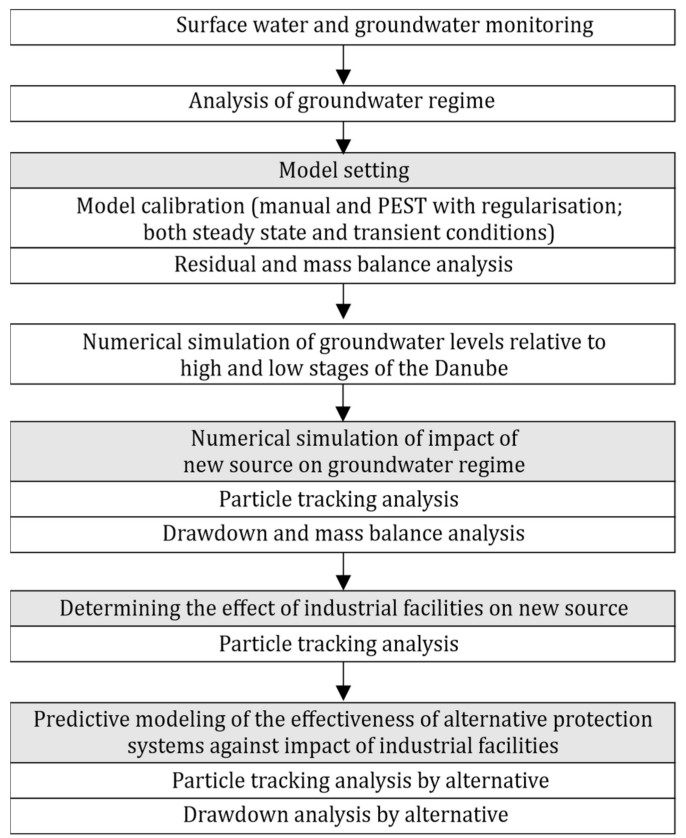

**Figure 3.** Investigations' algorithm.

The first task of the problem analysis comprises on-site activities and investigations, including data collection for the subsequent hydrodynamic/numerical modeling [33]. In hydrogeology, on-site investigations are undertaken to determine the lithologic composition, rock relationships and thicknesses, and their hydrogeologic and hydraulic functions. Special attention is devoted to observation structures, where the groundwater levels are measured. Hydrometeorological and hydrological methods are among the most often used and are virtually unavoidable in hydrogeology. The data are synthesized to define both the water circulation cycle and various elements, such as precipitation, surface runoff, inflow and evapotranspiration, or, in other words, all of the important parameters that directly quantify the aquifer recharge-effective infiltration. The quantity is an important factor when this parameter is entered into the hydrodynamic/numerical model.

The hydrodynamic (HD) analysis is a method for gaining insight into the groundwater regime in the study area and then examining the flow path and velocity. The HD analysis is a set of HD calculations. Today, the most comprehensive and commonly used method is a three-dimensional (3D) modeling of the aquifer regime, based on the numerical solving of differential equations that describe the groundwater flow and associated processes in the porous environment.

The groundwater flow in an unconfined aquifer may be approximately modeled by the nonlinear Boussinesq equation, assuming that Dupuit's hypothesis of horizontal flow applies [34]. This equation in the Cartesian coordinate system is:

$$\frac{\partial}{\partial x}\left(K_x \frac{\partial h}{\partial x}\right) + \frac{\partial}{\partial y}\left(K_y \frac{\partial h}{\partial y}\right) + \frac{\partial}{\partial z}\left(K_z \frac{\partial h}{\partial z}\right) - W = S\frac{\partial h}{\partial t} \tag{1}$$

where x, y and z are coordinates of the Cartesian coordinate system; $K_x$, $K_y$ and $K_z$ are the hydraulic conductivities along the x, y and z coordinates, which are assumed to be parallel to the major axes of hydraulic conductivity ($ms^{-1}$); h is the hydraulic head (m); W is the unit precipitation (precipitation per unit of horizontal spread of the flow), representing the effective intensity of vertical recharge, $[ms^{-1}]$; and S is the storage coefficient (-).

The HD modeling comprises several steps, in the following order: (i) examine the problem to be solved (purpose and goal of modeling); (ii) construct a hydrogeologic (HG) model; and (iii) convert the HG model into an HD model (select the numerical method and modeling software, discretize the area, import the model geometry, import the HG parameters of the porous environment, assign the boundary conditions of the model, specify the initial conditions of the model, select and discretize the computation interval, calibrate the HD model, analyze the HD model's sensitivity, validate the HD model, make alternative prognostic HD calculations, select the best alternative, update or recalibrate the HD model, and display the results). There are many literature sources that discuss the theoretical background and numerical modeling concept [35–37]. Groundwater Vistas Premium version 8.10 b.4 [25] and MODFLOW [38] were used to simulate the groundwater regime.

Calibration is a specific and important modeling step. It is a stage in which the simulation results (e.g., the aquifer regime) are matched to the data recorded on site (piezometric head and groundwater budget components). This is a sensitive and grueling process, which requires a lot of time to be successful. The model calibration can be manual or automatic, using special programs. Manual calibration is based on trial and error, using various combinations of the model parameter values. This approach requires extensive experience of the model developer and the question whether the solution reflects an optimal set of parameters remains open. A more efficient approach is needed, to avoid the subjective component and because of the large number of possible combinations of parameters. Such an approach is automated calibration, which optimizes the values of the selected parameters of the HG system. The underlying criterion, or target function, is to minimize the differences between the observed values and the results of simulation [39]. The PEST program [40], based on the Gauss-Marquardt-Levenberg algorithm, was used in this research for optimization. The concept requires the introduction of pilot points [41]. The pilot points need not be imaginary, but points at which some of the parameters are known are relatively rare. Each pilot point represents a parameter whose value needs to be determined. The concept of pilot points is not limited to hydraulic conductivity; it encompasses all types of parameters determined by model calibration. There are several ways in which pilot points can be assigned, such as a regular grid, target triangulation, target locations or fill gaps. The pilot points in the PEST require regularization of the distribution of certain parameters and/or values of the boundary conditions. Such an approach to PEST calibration produces a result that reflects the heterogeneity of the site (flow field), relative to the distribution of the hydraulic parameters that were determined.

In the HD analysis of groundwater flow, modeling of the transport of an ideal, conservative particle is the most commonly used method for determining the flow path and travel time of a pollutant in groundwater. This method disregards the dispersion effect and assumes no interaction between the particle and groundwater, or between the particle and the aquifer matrix. The direction of the pollutant migration is determined relatively quickly, as is the travel time from the pollution source to the pumping well. Groundwater Vistas Premium version 8.10 b.4 [25] and MODFLOW MODPATH v.7 [26] were used to analyze the flow path and residence time of an ideal particle.

## 4. Results and Discussion

The hydrogeological investigations referred to in this paper included the installation of a network of observation wells, monitoring of the groundwater levels in the study area, and gauging of the stages of the Danube and the Dunavac twice a month in 2015. The precipitation data were obtained from the National Hydrometeorological Service of Serbia.

A 3D HD model of the study area, based on finite differences, was constructed for the HD analysis of the groundwater regime, using MODFLOW 2005 [38].

The HD model of the extended area of the Jagodica source was developed as a multi-layer model, with four layers along the vertical. Each of the layers corresponded to a real stratum, schematized and identified based on knowledge of the terrain and analysis of the results of in situ investigations (Table 1).

**Table 1.** Representation of the flow field-lithologic units in the model layers with initial values of hydraulic conductivity (K), specific storage ($S_s$) and specific yield ($S_y$).

| Model Layer | Lithologic Units | $K_{x,y}$ (m/s) | $S_s$ (1/m) | $S_y$ (-) |
|---|---|---|---|---|
| First confining layer | Quaternary sandy clay, clayey sand and clay | $3 \times 10^{-7}$ | $1.15 \times 10^{-3}$ | 0.02 |
| Second water-bearing layer | Alluvial sands | $3 \times 10^{-4}$ | $1.2 \times 10^{-4}$ | 0.16 |
| Third water-bearing layer | Alluvial coarse gravel and sandy gravel | $3 \times 10^{-3}$ | $2.9 \times 10^{-5}$ | 0.23 |
| Fourth confining layer | Pliocene clay, sandy clay and silty clay | $1 \times 10^{-7}$ | $1.15 \times 10^{-3}$ | 0.02 |

The hydraulic parameters of the extended area of Jagodica are represented by the hydraulic conductivities and storage parameters of the lithologic units. The initial values of hydraulic conductivity (horizontal component—$K_{x,y}$) were assigned to each lithologic unit, based on earlier hydrogeologic investigations. The initial value of the vertical component was $K_z = 0.5\ K_{x,y}$. The storage parameters represented by specific yield ($S_y$) and specific storage ($S_s$) were specified according to the literature sources [42–48] (Table 1).

The basic dimensions of the study area matrix were 7800 m × 4200 m. The flow field in plan view was generally discretized with 100 m × 100 m cells. In parts of particular interest, the density was higher and the cell size was 25 m × 25 m. There were 98,468 active model cells. Figure 4 shows the basic model matrix and the discretization of the flow field in the extended area of Jagodica.

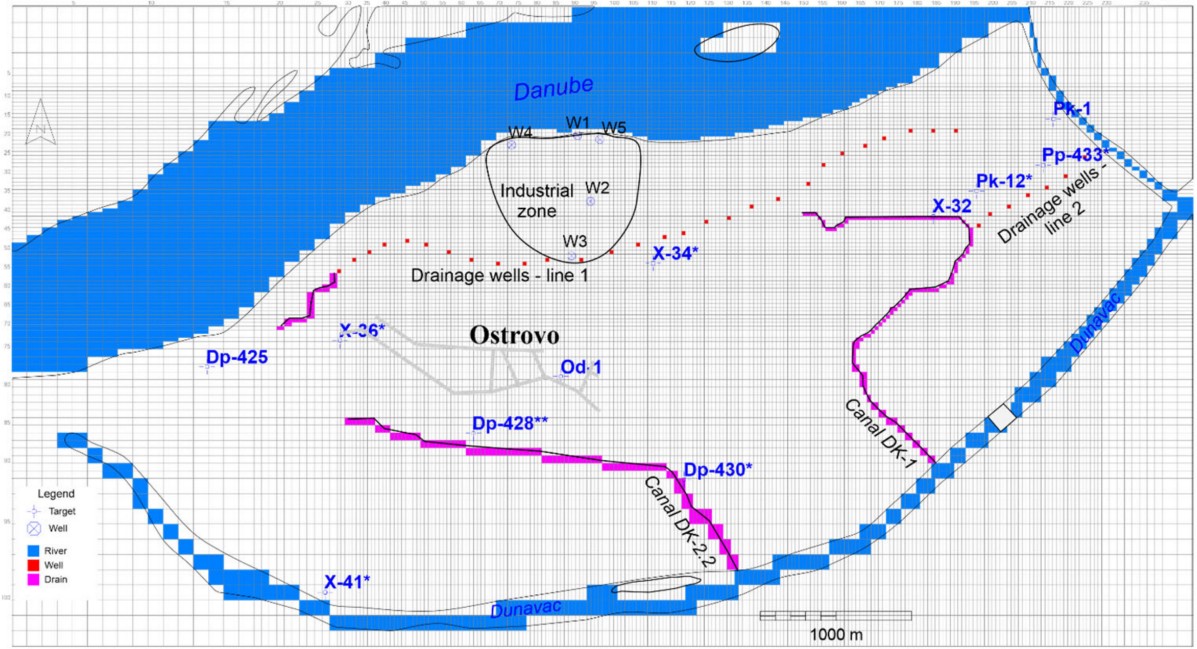

**Figure 4.** Discretization of the modeled study area and boundary conditions of the second model layer.

The following boundary conditions (BC) were assigned to the HD model of the extended area of Jagodica: effective infiltration; river, drain and prescribed flux.

*Effective infiltration.* Vertical balance plays an important role in the groundwater budget of the study area. Vertical balance is the effective/resulting infiltration, which is the sum of infiltrated precipitation, evaporation from the water table, and evapotranspiration. The depth to groundwater, moisture and lithologic composition of the overlying strata are also important. The daily precipitation totals recorded at a nearby rain-gauging station of the National Hydrometeorological Service of Serbia from 1 January to 31 December 2015 (Figure 5) were used to quantify the effective infiltration.

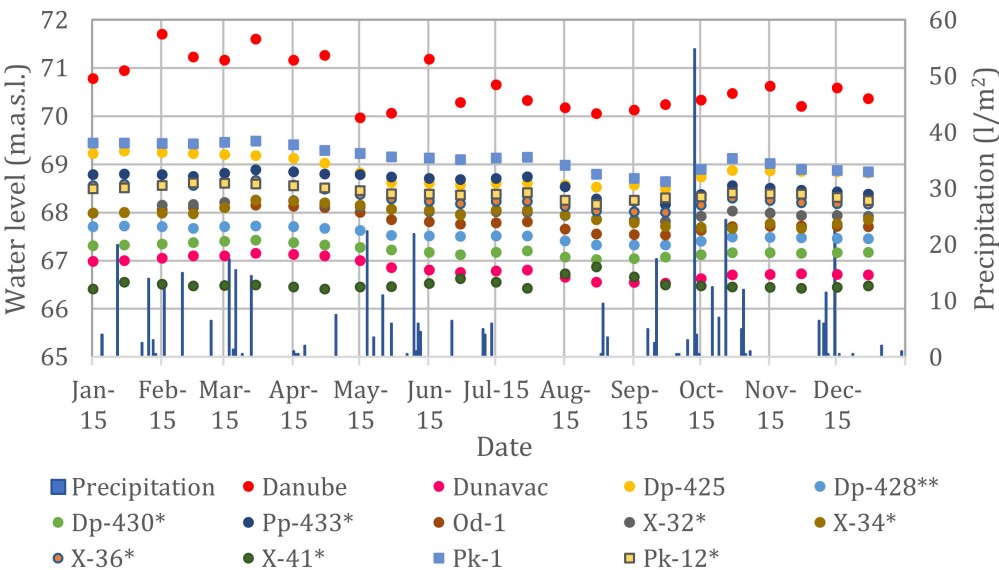

**Figure 5.** Histogram of daily precipitation totals, groundwater levels of the alluvial aquifer, and stages of the Danube and the Dunavac from 1 January to 31 December 2015.

The average effective infiltration was specified as a percentage of the mean monthly precipitation. The initial value was 12% of the precipitation. This boundary condition was assigned only to the first model layer.

*Boundary condition "river".* Streamflow, primarily of the Danube River, played the main role in defining the groundwater regime of the extended area of Jagodica. Figure 5 shows the stages of the Danube in 2015. In addition, the groundwater regime was affected by the Dunavac arm, whose water levels were governed by the operation of the Kolište II pumping station (Figure 1). The influence of streamflow was simulated by the BC River. The flow direction between the river and the aquifer depends on the groundwater level in a model cell, which is a result of calculations and the set water level of the river. This boundary condition was assigned to the second model layer (Figure 4).

*Boundary condition "drain".* This boundary condition simulated the influence of drainage canals. It was specified in three zones, or three groups of model cells, relative to the canal location, water level, elevation of the bottom of the model cell in which this boundary condition was specified, and the filtration properties of the related sediments. The value of conductivity for this boundary condition was one of the results of model calibration. The width and length of the drainage canal were specified relative to its geometry and the size of model cells. BC Drain was assigned to the sandy sediments/second model layer (Figure 4).

*Boundary condition "specified flux".* This boundary condition was used to simulate the operation of 35 drainage wells (individual capacity 3–3.5 L/s), whose function is to provide protection against high groundwater levels. There are two lines of drainage wells, one relatively close to the Danube and the other east of canal DK-1. The flow rates of the drainage wells were specified as constant values, in the second and third model layers. Figure 4 shows the locations in the second model layer.

Calibration was manual and automatic, using the PEST program (Doherty, 2010). The model was calibrated for both steady and unsteady conditions. Manual calibration was used for steady conditions, until a rough match between the modeled and observed groundwater levels was achieved. The model was calibrated for two points in time, when the stages of the Danube were at their maximum and minimum. Automatic calibration, using the PEST program with a regularization option, was applied to steady conditions at the minimum stages of the Danube, to determine the representative hydraulic conductivity of the aquifer. The latter involved the setting of pilot points, which yielded heterogenous zones with values of hydraulic parameters. The pilot points provided the spatial distributions of the horizontal ($K_x = K_y$) and vertical ($K_z$) components of hydraulic conductivity in the second and third model layers. The total number of pilot points was 264, with horizontal and vertical components of hydraulic conductivity, in the form of a homogeneous grid of pilot points whose dimensions were $\Delta x = \Delta y = 500$ m. After a satisfactory match was achieved between the modeled and observed groundwater levels at the minimum stages of the Danube, the groundwater regime at maximum stages was simulated effectively.

The unsteady calibration of the model was performed during the period of groundwater monitoring from 1 January to 31 December 2015. The basic time step of one day was consistent with the recording schedule of the groundwater budget components. At a lower iteration level, the one-day time step comprised ten parts of unequal duration (factor 1.2).

Due to a lack of exact data, the storage parameters were assigned homogeneously to each model layer. The representative values of this parameter were determined by using the conventional PEST program after steady calibration of the model.

During model calibration, the target points of the flow field were primarily the groundwater levels recorded at the observation wells. The monitoring network was comprised of 11 observation wells, whose screens were installed in gravel (third model layer). The locations of the observation wells are shown in Figure 4. The recorded groundwater levels used for model calibration are shown in Figure 5.

The calibration ended when a satisfactory match was achieved between the observed and modeled groundwater levels, including checking of the modeled groundwater budget. Figure 6 shows the groundwater levels in the water-bearing gravels at the highest stage of the Danube recorded in 2015, and Figure 7 shows the water table of the alluvial aquifer at the lowest stage of the Danube.

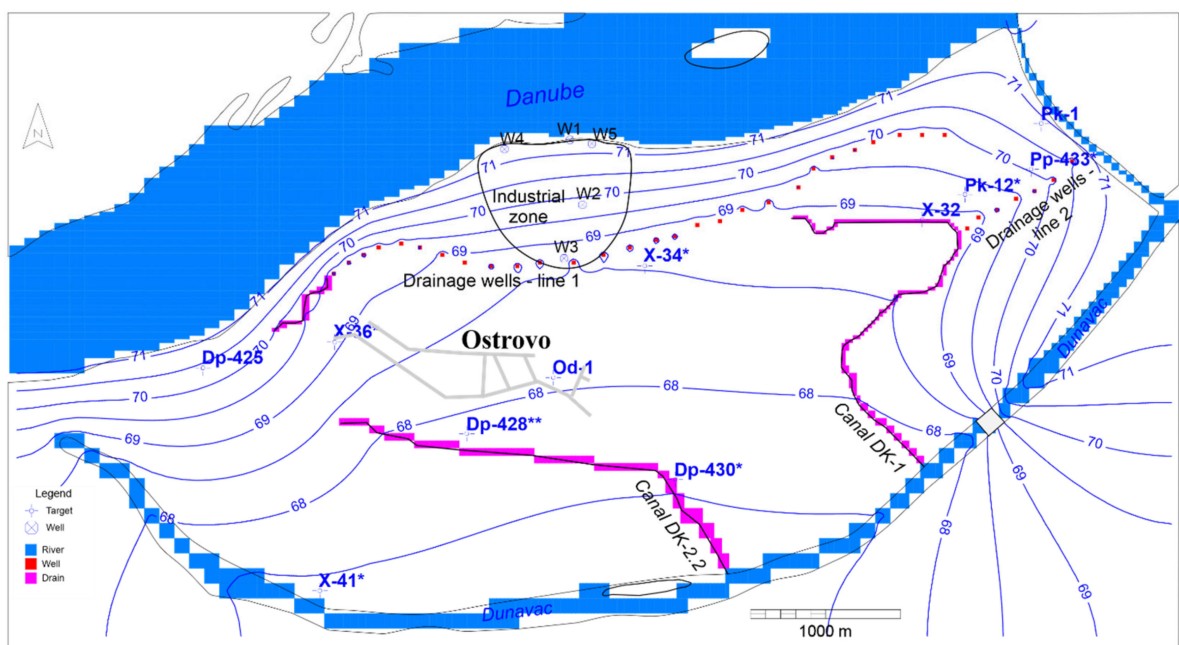

**Figure 6.** Water level distribution of the second model layer (sandy sediments) at high stages of the Danube.

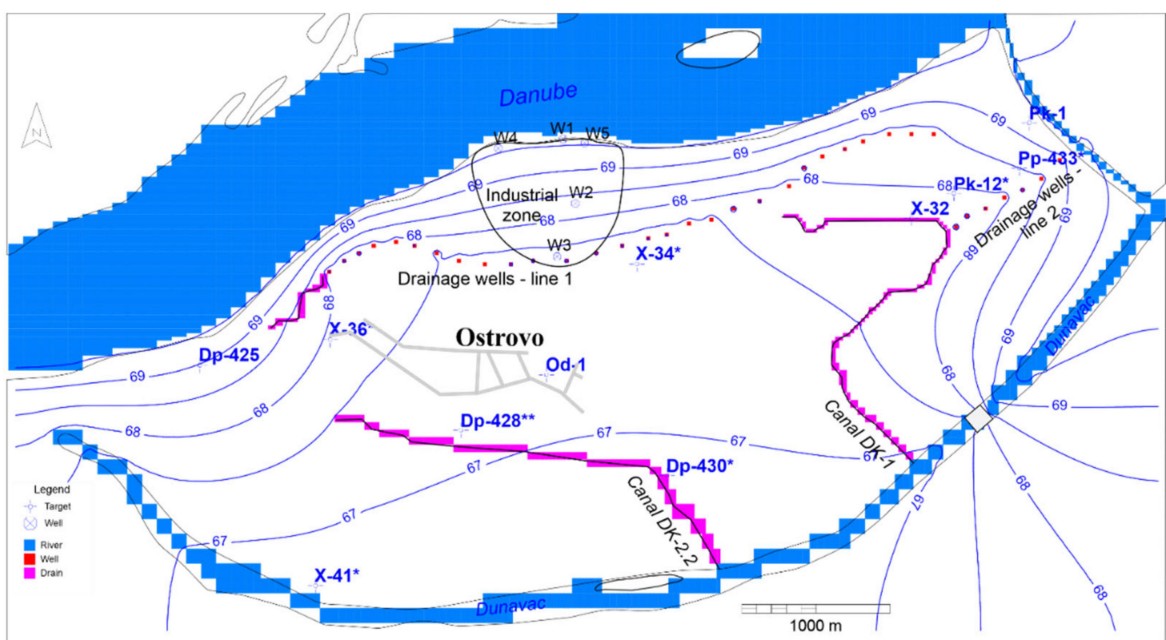

**Figure 7.** Water level distribution of the second model layer (sandy sediments) at low stages of the Danube.

Figures 8 and 9 show the observed and modeled groundwater levels at the ten observation wells from 1 January to 31 December 2015.

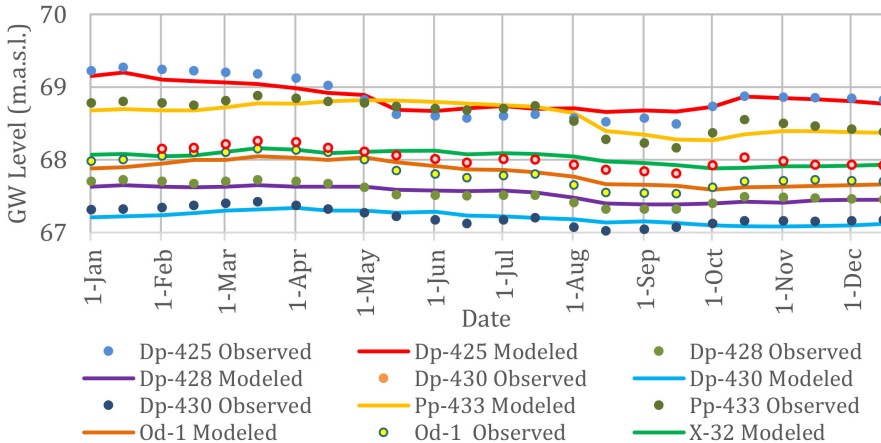

**Figure 8.** Observed and modeled groundwater levels at observation wells Dp-425, Dp-428, Dp-430, Pp-433, Od-1 and X-32 during the studied period (1 January to 31 December 2015).

The plots show a generally good match between the observed and modeled groundwater levels (residuals) at the observation wells. The groundwater budget of the study area is presented relative to the modeled boundary conditions at the high and low stages of the Danube in 2015. The Danube contributes 97–98% to the alluvial aquifer's recharge, depending on the river stage. The drainage system (drainage wells and canals) collects 312.5 L/s, or 68% of the groundwater that exits the model, regardless of the Danube's stage. The remaining 32% of that groundwater is drained into the Dunavac.

Hydraulic conductivity matrices ($K_{x,y}$) of the second and third model layers were among the results of calibration, as shown in Figures 10 and 11.

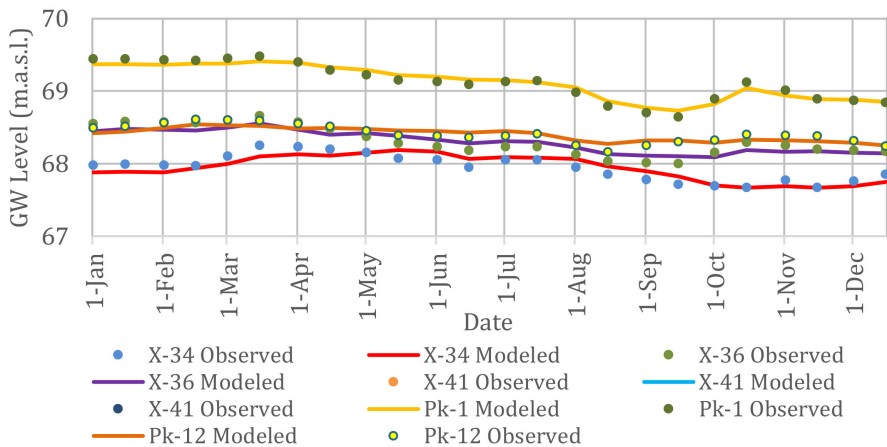

**Figure 9.** Observed and modeled groundwater levels at observation wells X-34, X-36, X-41, Pk-1 and Pk-12 (1 January to 31 December 2015).

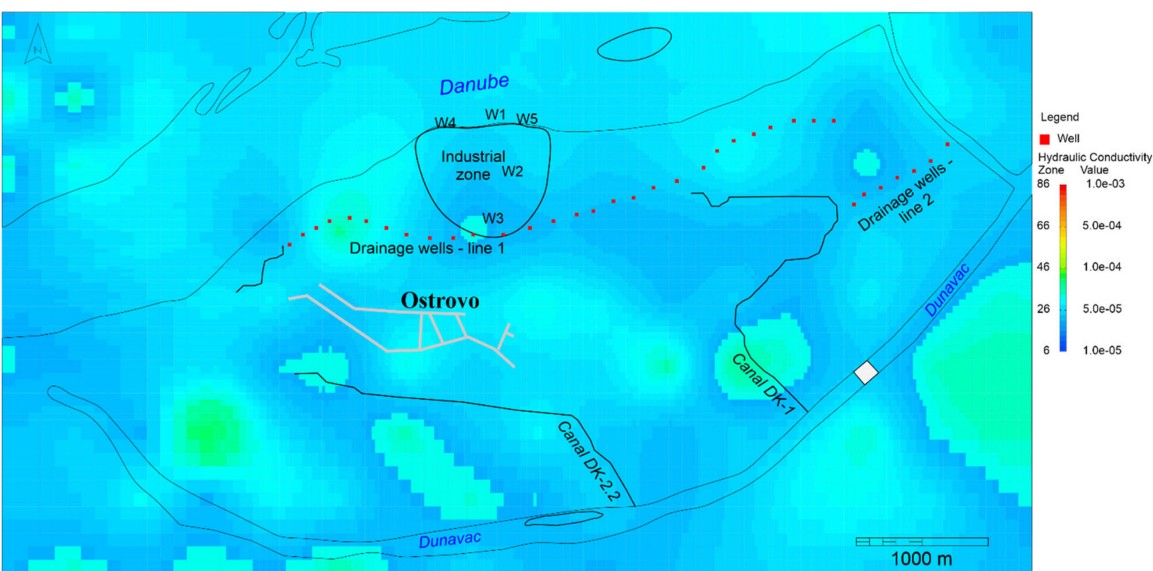

**Figure 10.** Zones and hydraulic conductivity $K_{x,y}$ matrix of the second model layer.

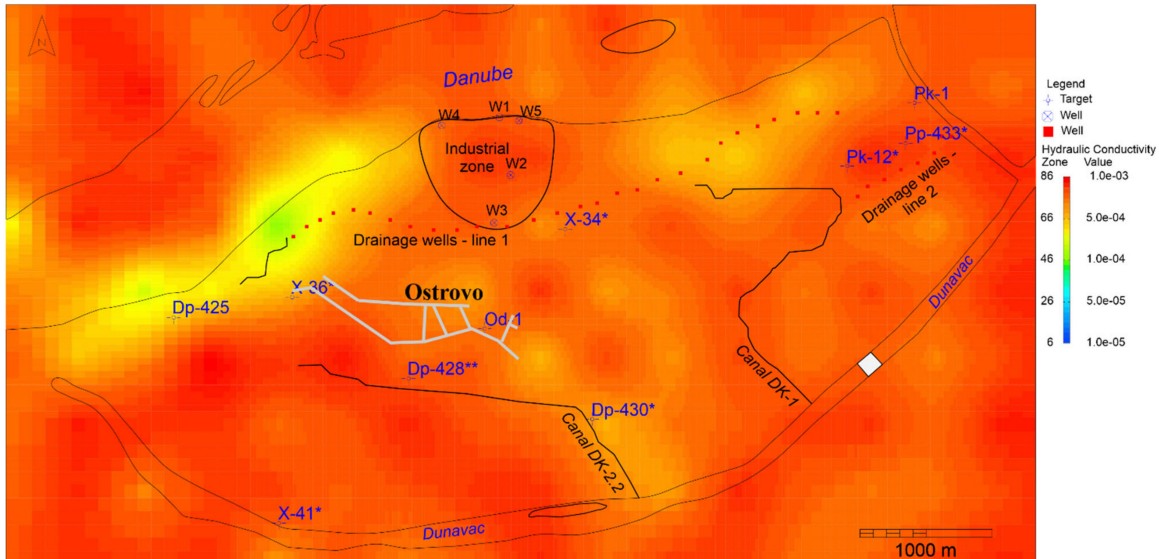

**Figure 11.** Zones and hydraulic conductivity $K_{x,y}$ matrix of the third model layer.

### 4.1. Forming of the Groundwater Source

The new groundwater source would be comprised of 20 pumping wells along the Danube, east of the industrial zone. The capacity of each well will be 20 L/s and the total capacity of the source will be 400 L/s. These wells were modeled with a grid of independent boundary conditions/analytical wells (Figure 12). The screens of the wells were assigned to the third model layer. When the source becomes operational, 17 drainage wells of the first drainage line will be decommissioned (Figure 12).

(**a**)

(**b**)

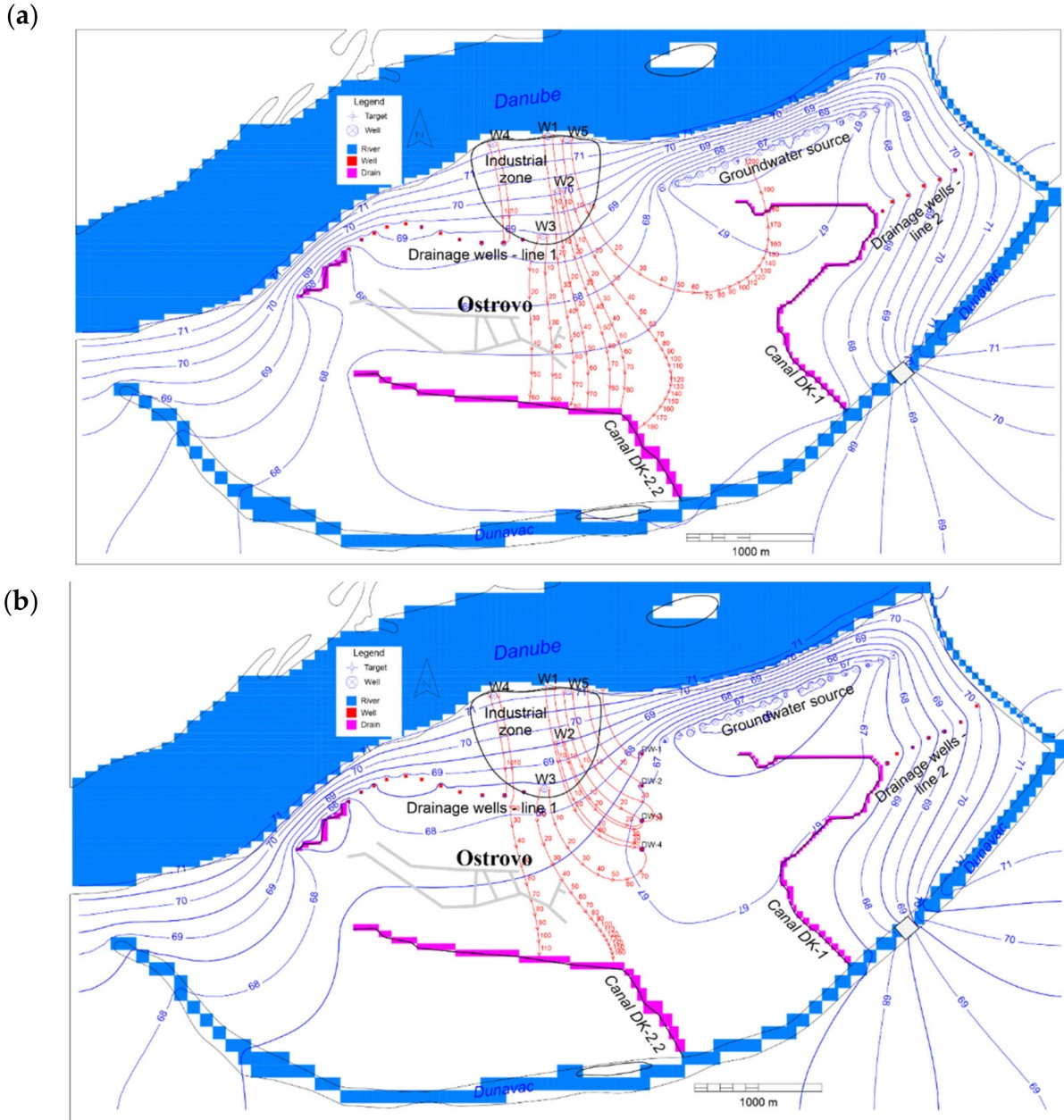

**Figure 12.** Hydroisohypses and recharge zone at (**a**) high and (**b**) low stages of the Danube in 2015 (second model layer).

Two cases were examined depending on the hydrology of the Danube and the Dunavac, reflecting the high and low stages in 2015. In each case, the migration of a conservative particle (tracer) was analyzed to determine the general direction of the groundwater flow from the industrial zone to the source, as well as the travel time to the pumping wells (Figure 12).

Figure 12 shows that the flow pattern differs from that which existed before the source was formed (Figures 6 and 7). With the source in place, the groundwater will be drawn from the extended area, including the industrial zone. The time needed for the groundwater from the industrial zone to reach the pumping wells will be 100 years at the high stages and 30–75 years at the low stages of the Danube. Such a long residence time is due to the significant role of the Danube in riverbank filtration within the immediate zone of the source.

When the Jagodica source becomes operational, the groundwater levels in its immediate zone will decline. At the highest recorded stages of the Danube and the Dunavac, the pumping wells will cause a drawdown of up to 3 m (Figure 13). The operation of Jagodica will not be "felt" in the industrial zone. There will be a larger drawdown at the low river stages, of 5 m at the source, 0.60 m at industrial zone W2 and W3, and <0.25 m at the other structures within the industrial zone.

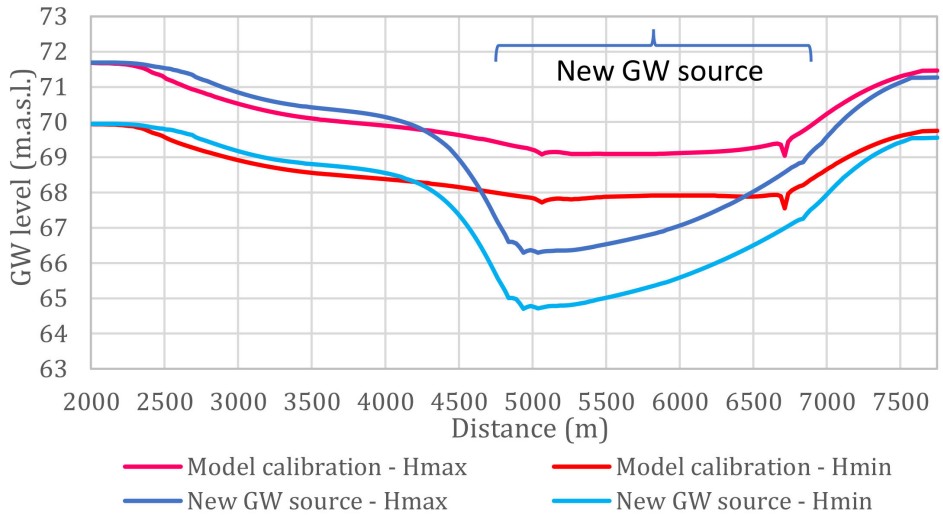

**Figure 13.** Groundwater level variations depending on the hydrology of the Danube before and after pumping wells become operational along the line shown in Figure 1.

In both of the hydrologic states, the Danube is the main source of "additional" aquifer recharge. Figure 14 shows the groundwater budgets at the low and high stages of the Danube recorded in 2015, when the Jagodica source was operational.

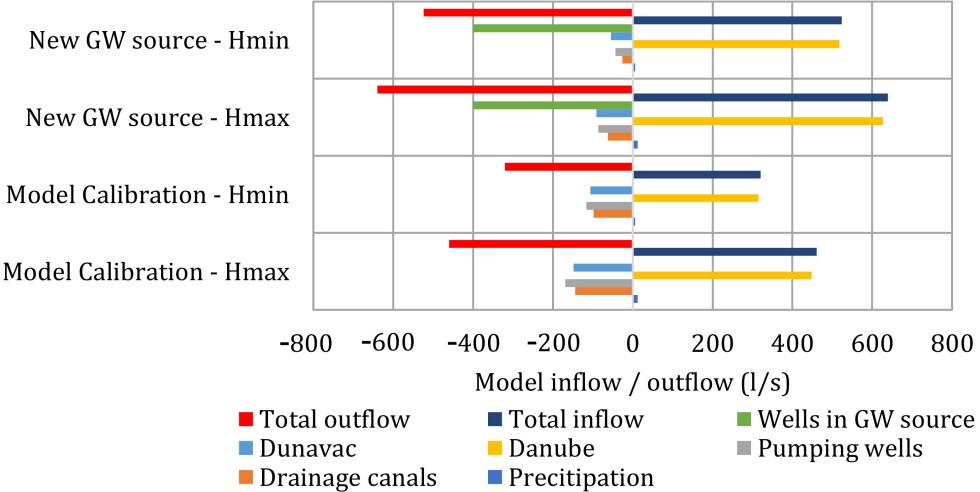

**Figure 14.** Groundwater budgets before and after the pumping wells become operational.

When the pumping wells are operated at 400 L/s, the riverbank filtration from the Danube will intensify, regardless of the stage. Compared to pre-existing conditions, the inflow from the Danube will be 40% greater in the wet period and 65% greater in the dry period. On the other hand, the inflow into the Dunavac will decline. This decline will be 38% at the high and 49% at the low stages of the Danube, compared to the pre-existing state. The operation of the pumping wells will clearly affect the drainage system as well. In addition to the reduced number of drainage wells, the flow to these wells will decrease relative to the pre-existing state. At the high stages of the Danube, the inflow will be 57% less, and the drainage canals will collect 49% less groundwater at the low river stages. At the low river stages, there will be 73% less inflow into the drainage canals and 63% less collection by the drainage wells than before the new source was created.

### 4.2. Groundwater Source Protection Management Plans

The effectiveness of the three protection systems/hydraulic barriers that would prevent the groundwater inflow from the industrial zone were analyzed. The components of the different systems would be in the same location. The main criteria for the selection of the location and characteristics of the systems were:

- prevention of flow to the pumping wells from the industrial zone and
- no major interference with recharge.

*Groundwater Source Protection Management Plan 1.* The hydraulic barrier would be a series of four drainage wells, with a north-east orientation (Figure 15). Their individual capacity would be 7.5 L/s. The function of these drainage wells would be to modify the flow pattern in the study area, or the drawing of groundwater from the direction of a (potential) pollution source and prevention of the pollutants from reaching the pumping wells. These wells were specified on the model by a constant flux boundary and the screens were emplaced in the sandy water-bearing layer (second model layer).

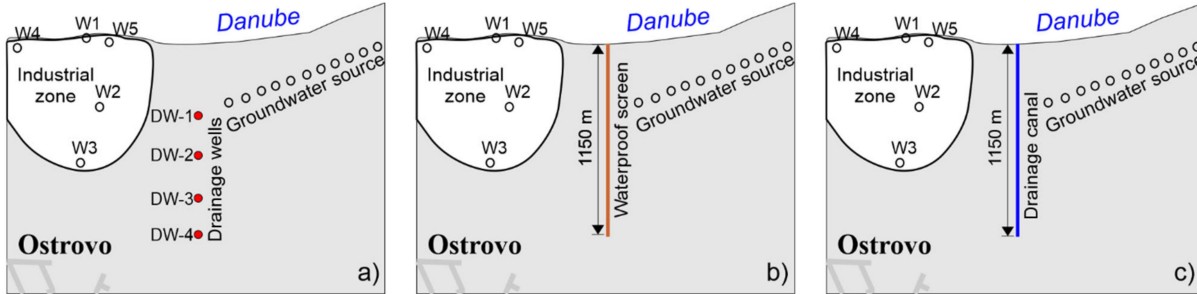

**Figure 15.** Schematic representation of the alternative plans. (**a**) Plan 1—Drainage wells; (**b**) Plan 2—Waterproof screen; (**c**) Plan 3—Drainage canal.

*Groundwater Source Protection Management Plan 2.* This hydraulic barrier would be an ideal (waterproof) screen. Its length would be 1150 m (Figure 15), width 1 m and depth 20 m (to the impermeable aquifer floor). One of the advantages of Plan 2 is reliability, given that the screen would be a fully impermeable, which the other alternatives are not.

*Groundwater Source Protection Management Plan 3.* This hydraulic barrier would comprise a drainage canal, whose geometry was determined by the prognostic HD simulations (Figure 5). The canal would be 1150 m long, 2 m wide and 10 m deep. The influence of the canal was simulated by a head-dependent boundary condition and the water level was specified as 1 m lower than that of the other drainage canals. The head-dependent boundary condition was assigned to the second model layer.

The calibrated model was used to predict the effectiveness of the alternative protection systems against inflow from the industrial zone, at the high and low stages of the Danube. The results are represented by the maps of the alluvial aquifer water table, streamlines and the travel time of the pollutant (simulated as a conservative particle) from the industrial facilities to the source protection system or drainage canal DK-2.2 (Figures 16–18).

**(a)**

**(b)**

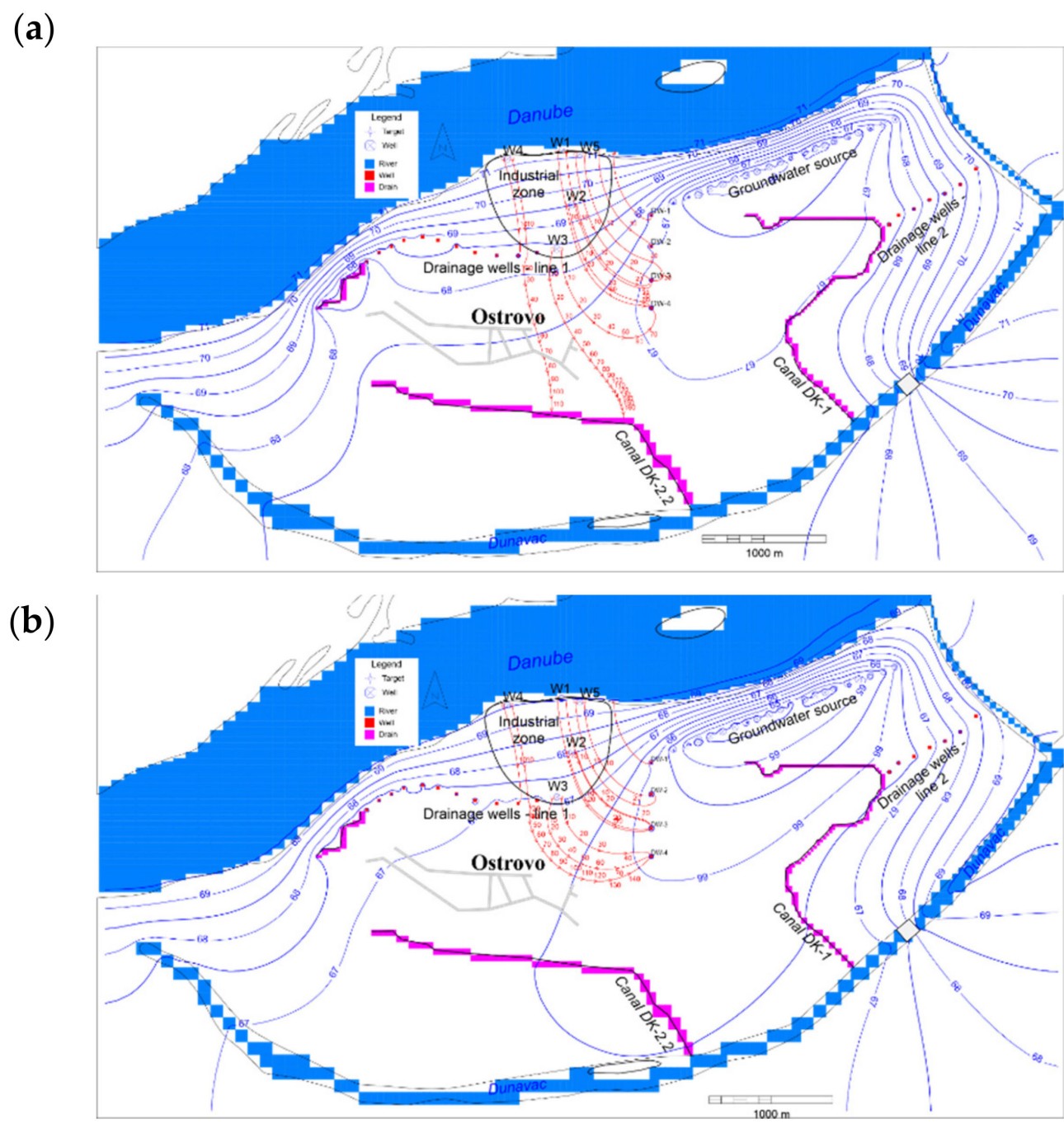

**Figure 16.** Flow pattern of Plan 1 (drainage wells) at the (**a**) high and (**b**) low stages of the Danube recorded in 2015 (second model layer).

Analysis of the flow path and the residence time of an inert particle, according to Plan 1, revealed that the drainage wells in the selected location meet the set criteria and would constitute a barrier for potential pollutants migrating from the industrial zone to the Jagodica source. Some of the advantages of drainage wells are low capital expenditure, quick and easy installation, and simple on/off operation. In addition, it would be possible to modify the capacity of the wells by installing different types of pumps, to provide flexibility. However, the wells are vulnerable to unscheduled shutdowns, for instance due to power outage. Other shortfalls of this approach include the need for labor to maintain the wells and pumps and to handle occasional repairs or replacement, as well as relatively high power consumption by the pumps. There is also an environmental impact due to

drawdown, the amount of pumped water and the problems associated with transport to a recipient. The groundwater budget, or flow to and from the hydrogeologic system, would also be affected.

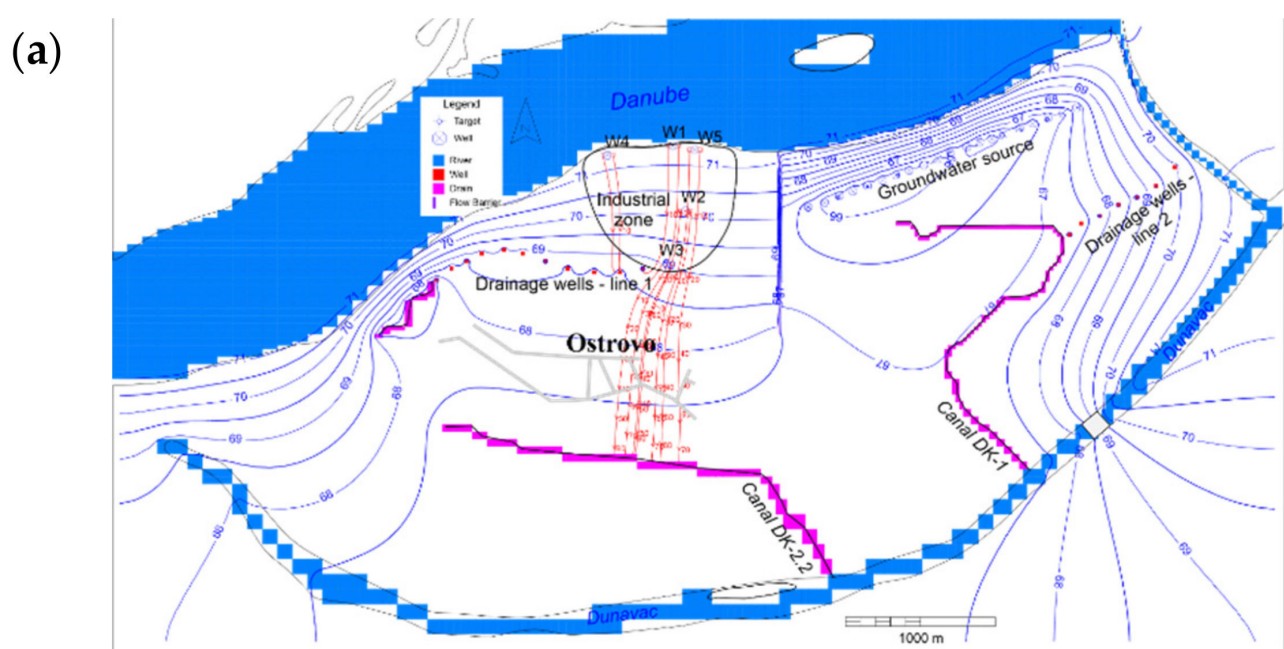

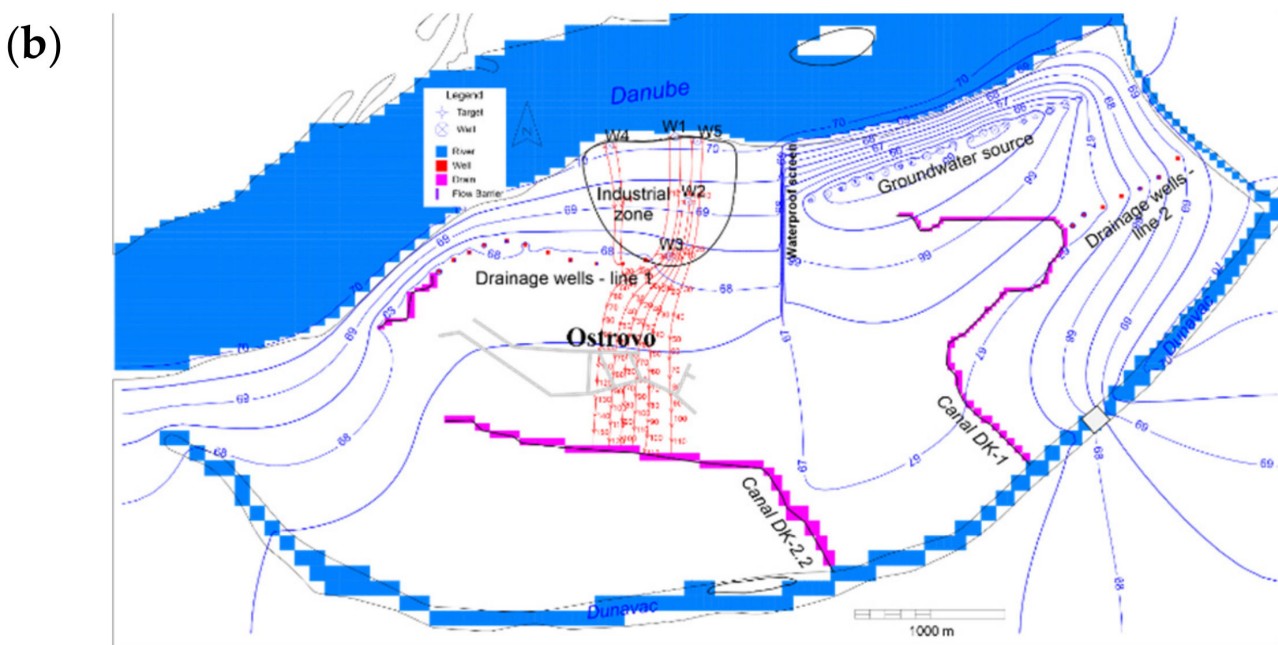

**Figure 17.** Flow pattern of Plan 2 (waterproof screen) at the (**a**) high and (**b**) low stages of the Danube recorded in 2015 (second model layer).

In Plan 2 (Figure 17), the flow path of an inert particle does not lead to the waterproof screen. Contrary to the drainage wells' scenario, the screen would not cause any drawdown, would not have an adverse effect on the environment, and would not require water evacuation. The screen does not need any electric power or maintenance and can be used in perpetuity. Some of the shortfalls of this approach are the special installation conditions, such as shallow depth or the presence of an impermeable horizon. These conditions exist

in the study area. In addition, waterproof screens require special installation technologies and machinery, resulting in high initial capital expenditure.

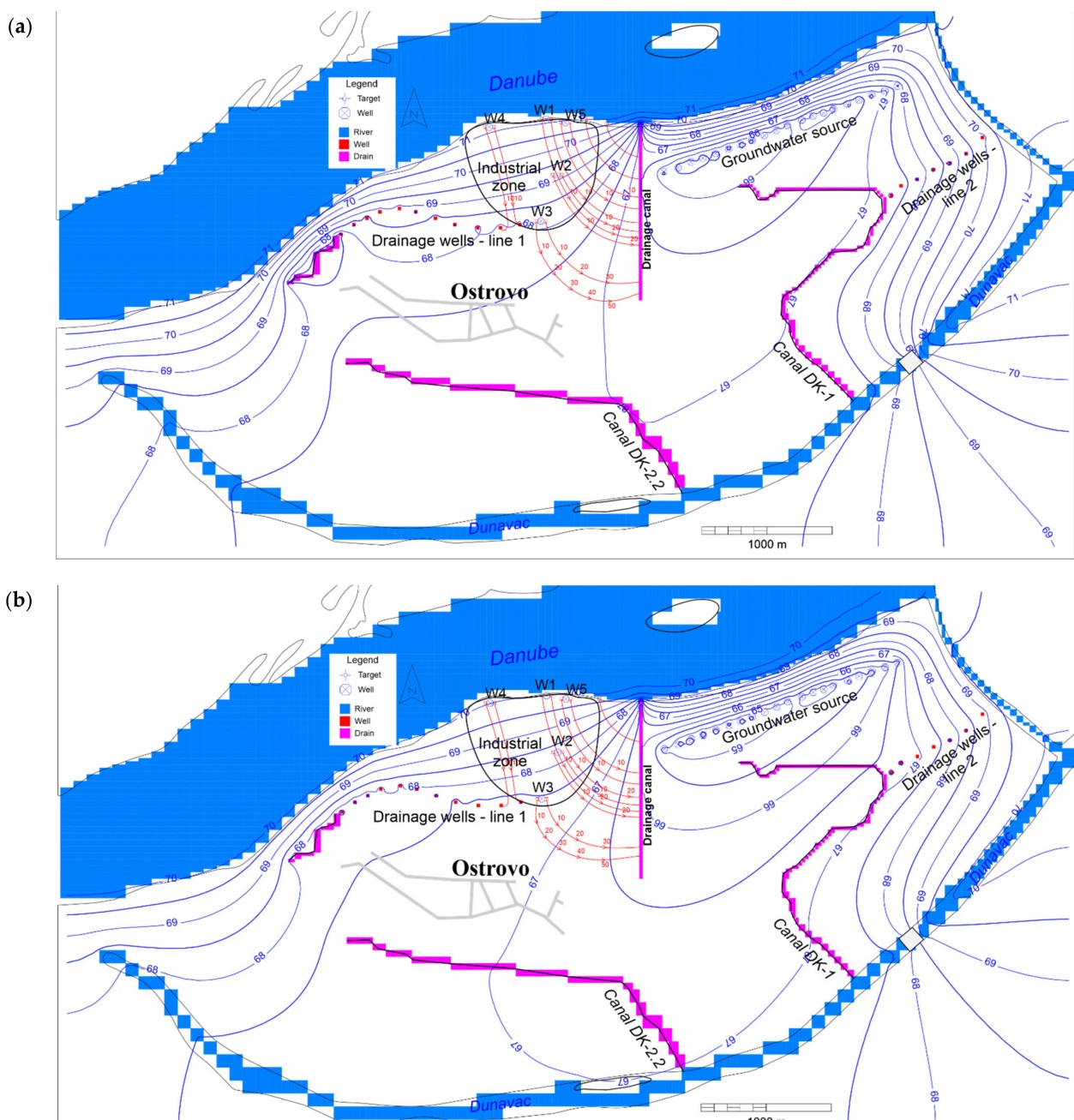

**Figure 18.** Flow pattern of Plan 3 (drainage canal) at the (**a**) high and (**b**) low stages of the Danube recorded in 2015 (second model layer).

In Plan 3, the drainage canal would be at the local base level, so the groundwater flow from the industrial zone would be towards the canal. It would be easy to dig the canal through the water-bearing sediments, which is an advantage over the other plans. Canals favor sediments comprised of gravel or sandy gravel, which is the case in the study area. The advantages of this plan over the other two are related to capital expenditure and any repair costs being associated only with periodic cleaning and maintaining slope stability and the hydraulic condition of the bottom of the canal. There are also environmental advantages, compared to Plan 1 for instance. In addition, there are technical benefits in

terms of simplicity, speed and low cost of emplacement, as well as efficiency. However, the canal alternative would require electric power for the pumping station that would control the water level in the canal. Figure 19 shows the effect of the protection systems on groundwater levels along section A–B (see Figure 2).

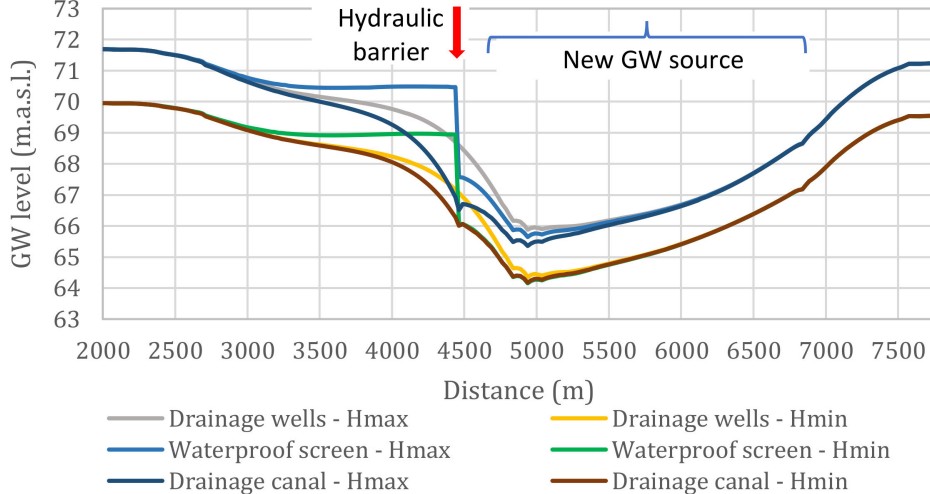

**Figure 19.** Groundwater levels along section A–B (see Figure 2) in the three studied pollution prevention scenarios.

Compared to the pre-existing conditions, Plans 1 and 3 would cause an additional drawdown, whereas Plan 2 (the waterproof screen) would raise the water table because there would be no influence of the pumping wells. At the high stages of the Danube, the additional drawdown caused by Plans 1 and 3 would be 0.41–1.10 m, and at low stages 0.36–0.63 m.

In the two considered hydrologic states of the Danube, the groundwater levels at the pumping wells would be virtually identical, regardless of the selected plan (Figure 19). The minimum levels would be slightly more uniform at the low stages of the Danube. The additional drawdown caused by the protection system would be 0.42–0.90 m at the high stages and 0.34–0.53 m at the low stages, depending on the type of system. However, none of the plans threatens the hydraulic function of the pumping wells, irrespective of the hydrology. Table 2 shows the travel time of a conservative particle from the waterproof screen.

**Table 2.** Travel time of conservative particle from industrial wells to source protection systems.

| Potential Pollution Source | Groundwater Source | | Plan 1 Drainage Wells | | Plan 2 Waterproof Screen | | Plan 3 Drainage Canal | |
|---|---|---|---|---|---|---|---|---|
| | Hmin | Hmax | Hmin | Hmax | Hmin | Hmax | Hmin | Hmax |
| W1 | - | - | 15–20 | 15–20 | - | - | 10–15 | 10–15 |
| W2 | 75–115 | - | 15–20 | 15–20 | - | - | 15 | 10–15 |
| W3 | - | - | 25–40 | 40 | - | - | 20–30 | 20–30 |
| W4 | - | - | 75 | - | - | - | - | - |
| W5 | 30–50 | 100 | 10–15 | 15 | - | - | 10 | 5–10 |

Table 2 shows that when the pumping wells are operational, a pollutant from industrial zone well W5 can reach the groundwater source, regardless of the stage of the Danube, and from W2 only at the low stages. If Plan 1 or 3 is selected, the groundwater from the industrial zone will flow to the protection system structures. In the case of Plan 2, the groundwater from the industrial zone would not reach the pumping wells.

At the low stages of the Danube, groundwater from W4 would reach the drainage wells in the case of Plan 1 (Figure 16). In all of the other cases, the groundwater will

gravitate toward the pre-existing drainage system. According to Plan 1, the travel time from the industrial zone to the drainage wells would be virtually the same—10 to 25 years—except from W3 and W4 where the travel time would be 25 to 40 years. In Plan 3, the groundwater from the industrial zone would reach the drainage canal in 5 to 15 years from W1, W2 and W5, or 20 to 30 years from W3, regardless of the Danube's stages.

The advantages and shortfalls of the protection management plans were analyzed based on the assumption that the system operates continuously and that all of its pollution prevention components are engaged throughout (Table 3). Each of the management plans was evaluated with respect to the capital expenditure, operating expenses, efficiency, vulnerability and environmental impact of the system.

**Table 3.** Advantages and shortfalls of the management plans.

|  | Capital Expenditure | Operating Expenses [1] | Efficiency of the System [2] | Vulnerability of the System [3] | Environmental Impact [4] |
|---|---|---|---|---|---|
| Plan 1 Drainage wells | Low | High | Moderate | High | Moderate |
| Plan 2 Waterproof screen | High | N/A | High | N/A | High |
| Plan 3 Drainage canal | Moderate | Moderate | Moderate | Moderate | High |

[1] Power consumption, periodic rehabilitation of system components, etc.; [2] Effectiveness of the system in operation (Figures 16–19); [3] Power outage, failure of a component, speed of replacement; [4] Consequences of drawdown: effect on vegetation, drying up of private shallow wells, etc.

## 5. Conclusions

The paper presented an investigative algorithm developed to quantify the potential pollution of a new groundwater source of public water supply and design prevention measures against a neighboring industrial zone. The first stage of the study involved monitoring and analyses of the surface water and groundwater regimes in the study area. The next step was a hydrodynamic analysis involving numerical modeling. In the specific case, the model was calibrated using the PEST program with a regularization option, which required specifying the pilot points. The calibrated model was used to simulate and assess the operating conditions of the pumping wells and their effect on the groundwater budget components and groundwater levels at the low and high stages of the Danube River.

The same model was used to determine the effect of potential pollution from the industrial zone on the pumping wells, through particle tracking and drawdown analysis. The study revealed that the pollutants from some of the industrial zone facilities might reach the pumping wells, so three groundwater source protection management plans were examined. Given that it was impossible to ascertain the potential source of pollution, pollutant migration was simulated by a conservative particle traveling from the ground locations of the industrial facilities.

Based on the flow path and travel time of a conservative particle in all three of the cases, the protection systems would meet the set criterion—to act as a hydraulic barrier for potential pollutants migrating from the industrial zone. There would be no significant impact on recharge. The selection of the best plan would require multicriteria optimization and analysis of the individual plans, which was beyond the scope of this study.

The contribution of this research is reflected in the possibility of future implementation of multicriteria decision making to select the optimal groundwater source protection management plans. This would involve numerical modeling, as described herein, and then mathematical optimization calculations to select the best plan. The proposed approach intertwines two different scientific disciplines, hydrodynamics and multicriteria optimization, and the research opens doors for a further interdisciplinary approach to the stated problem.

Moreover, the contribution of the proposed methodology is not limited to the impact of an industrial zone on a groundwater source. It is also applicable to other cases that involve

sources of pollution and prevention of groundwater source contamination (excessive pollution, makeshift landfills, tailings' disposal sites, etc.), as well as the conservation of recharge zones (e.g., due to urban development in relative proximity to water wells), maintenance of design groundwater levels to conserve the hydraulic function of pumping wells, and similar instances.

**Author Contributions:** All of the authors jointly contributed to the finalization of the paper: D.P. (Dušan Polomčić) and D.B. designed the numerical model and alternatives; B.H. carried out initial data analysis; D.P. (Dragan Pamučar) designed the research methodology and critically reviewed the concept and design of the paper. All authors have read and agreed to the published version of the manuscript.

**Funding:** This research received no external funding.

**Institutional Review Board Statement:** Not applicable.

**Informed Consent Statement:** Not applicable.

**Data Availability Statement:** Not applicable.

**Acknowledgments:** The authors express their gratitude to the Ministry of Education, Science and Technological Development of the Republic of Serbia for supporting scientific research, which is essential for the advancement of a knowledge-based society.

**Conflicts of Interest:** The authors declare no conflict of interest. The funders had no role in the design of the study; in the collection, analyses, or interpretation of data; in the writing of the manuscript, or in the decision to publish the results.

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
