# Peer review of "Numerical Modeling and Simulation of the Effectiveness of Groundwater Source Protection Management Plans: Riverbank Filtration Case Study in Serbia"

_water, doi:10.3390/w14131993_

Round 1

Reviewer 1 Report

The paper is well-conceived and written.

The figures are pretty straightforward.

However, I disagree with one aspect. Even in the conclusions, the authors say they have proposed a method in the paper. On this aspect, I can't entirely agree. The results of applying a numerical model under varying boundary conditions are reported and analysed. It would be appropriate for the authors to specify what innovative contribution they make with this paper.

Author Response

REV 1

Comments and Suggestions for Authors

The paper is well-conceived and written.

The figures are pretty straightforward.

However, I disagree with one aspect. Even in the conclusions, the authors say they have proposed a method in the paper. On this aspect, I can't entirely agree. The results of applying a numerical model under varying boundary conditions are reported and analysed. It would be appropriate for the authors to specify what innovative contribution they make with this paper.

RESPONSE: Agree. Changes made. Manuscript revised accordingly.

Reviewer 2 Report

The paper is interesting. The structure of the paper is correct. Nevertheless, some aspects of the paper should be revised and improved:

- The authors should review keywords.

- In the introduction, cases of groundwater pollution are reviewed. Some computer programs are also reviewed. However, a comprehensive review of the research topic of the paper is lacking.

- A research paper must be on the frontier of knowledge. What is the scientific novelty of the work carried out? The introduction should be clear about the novelty of the work presented compared to other works already published.

- In the work presented, several computer programs are used to solve a problem of groundwater contamination, but I am not clear about the novelty of the research. In my opinion, the presented work is a case study rather than a research paper.

- Lines 179-180: "(i)" and "(ii)" are missing.

- Calibration of any mathematical model is very important for the model to be useful. The authors should explain in much more detail the calibration process.

- What parameters are modified in the calibration process?

- It would be interesting if the advantages and disadvantages of the three plans appeared in a table.

- The “Abstract” section is a summary of the paper, but the "Conclusions" section should not be a summary of the paper. In this section, the authors must specify the conclusions of the research. The authors should complete and improve the "Conclusions" section.

To conclude, in my opinion, the paper needs major modifications to be published in a prestigious scientific journal. The authors must clarify the scientific novelty of the work carried out. On the other hand, the authors must complete and improve some sections.

Author Response

REV 2

The paper is interesting. The structure of the paper is correct. Nevertheless, some aspects of the paper should be revised and improved:

- The authors should review keywords.

RESPONSE: Agree. Changes made. Manuscript revised accordingly.

- In the introduction, cases of groundwater pollution are reviewed. Some computer programs are also reviewed. However, a comprehensive review of the research topic of the paper is lacking.

RESPONSE: Agree. Changes made. Manuscript revised accordingly.

- A research paper must be on the frontier of knowledge. What is the scientific novelty of the work carried out? The introduction should be clear about the novelty of the work presented compared to other works already published.

RESPONSE: Agree. Changes made. Manuscript revised accordingly. Also, in the conclusion, the facts of the contribution of this paper are written in more detail.

- In the work presented, several computer programs are used to solve a problem of groundwater contamination, but I am not clear about the novelty of the research. In my opinion, the presented work is a case study rather than a research paper.

RESPONSE: Agree. Changes made. Manuscript revised accordingly. Also, we agree that this is a case study. In that context, we also changed the title of the paper.

- Lines 179-180: "(i)" and "(ii)" are missing.

RESPONSE: Agree. Changes made. Manuscript revised accordingly.

- Calibration of any mathematical model is very important for the model to be useful. The authors should explain in much more detail the calibration process.

RESPONSE: Agree. Changes made. Manuscript revised accordingly.

- What parameters are modified in the calibration process?

RESPONSE: Agree. Changes made. Manuscript revised accordingly.

- It would be interesting if the advantages and disadvantages of the three plans appeared in a table.

RESPONSE: Agree. Changes made. Manuscript revised accordingly.

- The “Abstract” section is a summary of the paper, but the "Conclusions" section should not be a summary of the paper. In this section, the authors must specify the conclusions of the research. The authors should complete and improve the "Conclusions" section.

RESPONSE: Agree. Changes made. Manuscript revised accordingly.

To conclude, in my opinion, the paper needs major modifications to be published in a prestigious scientific journal. The authors must clarify the scientific novelty of the work carried out. On the other hand, the authors must complete and improve some sections.

RESPONSE: Agree. Changes made. Manuscript revised accordingly.

Round 2

Reviewer 2 Report

The modifications made to the manuscript are very small. The authors have only added a few lines, but the manuscript is practically identical to the first version. In addition, some comments have not been taken into account, for example:

- In the introduction, cases of groundwater pollution are reviewed. Some computer programs are also reviewed. However, a comprehensive review of the research topic of the paper is lacking.

- Lines 188-190: "(i)" and "(ii)" are missing.

- What parameters are modified in the calibration process?

- The “Abstract” section is a summary of the paper, but the "Conclusions" section should not be a summary of the paper. In this section, the authors must specify the conclusions of the research. The authors should complete and improve the "Conclusions" section.

The paper must be completed and improved.

Author Response

REV 1

The modifications made to the manuscript are very small. The authors have only added a few lines, but the manuscript is practically identical to the first version. In addition, some comments have not been taken into account, for example:

- In the introduction, cases of groundwater pollution are reviewed. Some computer programs are also reviewed. However, a comprehensive review of the research topic of the paper is lacking.

RESPONSE: RESPONSE: Agree. Changes made. Manuscript revised accordingly. The essence is described in lines 108-114. As written in the Introduction, we would like to emphasize that different procedures are used to protect groundwater from pollutants, and this paper is specific in that it used numerical analysis and groundwater modeling to create different defense plans.

- Lines 188-190: "(i)" and "(ii)" are missing.

RESPONSE: Agree. Changes made. Manuscript revised accordingly.

- What parameters are modified in the calibration process?

RESPONSE: Agree. Changes made. Manuscript revised accordingly. The parameters considered are the following:

- hydraulic conductivity (Kx, y and Kz in the second and third model layers (line 307);

- storage parameters (storage parameters were assigned homogeneously to each model layer line 318)

Aslo, everything is shown in the Figures.

- The “Abstract” section is a summary of the paper, but the "Conclusions" section should not be a summary of the paper. In this section, the authors must specify the conclusions of the research. The authors should complete and improve the "Conclusions" section.

RESPONSE: Agree. Changes made. Manuscript revised accordingly. In the Conclusion, the parts related to the results and discussion of the analyzed case study were deleted. A section on scientific contributions has been added.

The paper must be completed and improved.

RESPONSE: Agree. Changes made. Manuscript revised accordingly. We also took into account the comments of other reviewers, so everything was harmonized.
